# Pathophysiology of Pulmonary Fibrosis in the Context of COVID-19 and Implications for Treatment: A Narrative Review

**DOI:** 10.3390/cells11162489

**Published:** 2022-08-11

**Authors:** Son Tran, Andre Ksajikian, Juliana Overbey, Patrick Li, Yong Li

**Affiliations:** 1Department of Orthopaedic Surgery, BioMedical Engineering, Western Michigan University Homer Stryker M. D. School of Medicine, Kalamazoo, MI 49008, USA; 2Stephen M. Ross School of Business, University of Michigan, Ann Arbor, MI 48109, USA

**Keywords:** pulmonary fibrosis, COVID-19, TGF-β1, prevention, treatment

## Abstract

Pulmonary fibrosis (PF) is a feared outcome of many pulmonary diseases which results in a reduction in lung compliance and capacity. The development of PF is relatively rare, but it can occur secondary to viral pneumonia, especially COVID-19 infection. While COVID-19 infection and its complications are still under investigation, we can look at a similar outbreak in the past to gain better insight as to the expected long-term outcomes of COVID-19 patient lung function. In the current article, we review the literature relative to PF via PubMed. We also performed a literature search for COVID-related pathological changes in the lungs. Finally, the paper was reviewed and summarized based on the studies’ integrity, relative, or power calculations. This article provides a narrative review that endeavors to elucidate the current understanding of the pathophysiological mechanisms underlying PF and therapeutic strategies. We also discussed the potential for preventing progression to the fibrotic state within the context of the COVID-19 pandemic. With the massive scale of the COVID-19 pandemic, we expect there should more instances of PF due to COVID-19 infection. Patients who survive severe COVID-19 infection may suffer from a high incidence of PF.

## 1. Introduction

Pulmonary fibrosis (PF) is a feared outcome of many pulmonary diseases as the progression of PF results in a reduction in lung compliance and capacity as well as an increase in alveolar wall thickness. These changes combine to result in poor lung function and labored respiration in afflicted patients. The mechanisms underlying PF are complex and currently not completely characterized due to its many etiologies. The development of PF is relatively rare, but it can occur secondary to viral pneumonia, especially COVID-19 [1]. With the massive scale of the COVID-19 pandemic, we expect there should more instances of PF due to COVID-19 infection.

While COVID-19 infection and its complications are still under investigation, we can look at a similar outbreak in the past to gain better insight as to the expected long-term outcomes of COVID-19 patient lung function. When compared to a similar infection, severe acute respiratory syndrome (SARS), which first appeared in China in 2002, some patients experienced a decline in respiratory function after the infection. A study looking at the pulmonary function of 97 patients who survived SARS in a one-year follow-up found that 28% had chest X-ray abnormalities. The pulmonary function testing shows that 4% of patients had a decreased forced vital capacity (FVC), 5% had a decrease in total lung capacity (TLC), and 24% had a decreased diffusion capacity for carbon monoxide (DLCO). A total of 32% of the patients that were admitted to the intensive care unit reported having a worse quality of life [2]. Overall, studies looking at SARS survivors and long-term health outcomes show 28% to 53% percent having decreased lung function similar to that of symptoms of pulmonary fibrosis [2,3,4]. This shows that post-COVID-19 PF is a significant concern.

In this viewpoint, we will review the current understanding of the pathophysiological mechanisms underlying PF and therapeutic avenues for the prevention of progression to the fibrotic state within the context of the COVID-19 pandemic. We will also seek to characterize patients who are the most at risk for the development of PF and targeted therapeutic approaches to improve their outcomes. Of note, patients with idiopathic pulmonary fibrosis (IPF) and patients with severe post-COVID-19-infection PF share illuminatingly similar patient characteristics. As such, we will examine the shared mechanisms behind the development of PF in both disease states. We first look at the pathophysiology behind IPF and then draw parallels to post-COVID-19 PF to allow us to better understand the progression of the disease.

## 2. Pathophysiology of Pulmonary Fibrosis

The pathophysiology of PF is an area of ongoing research. This disease state can result from many etiologies which may result in different primary mechanisms that drive the shared fibrotic outcome. Patients who develop acute respiratory distress syndrome (ARDS) [5] may experience in an excessive proliferation of fibrotic factors leading to pulmonary fibrosis. Other patients may develop fibrosis from environmental insults, such as chest irradiation or chronic occupational exposure to irritants such as asbestos or silica. Rarely, patients may develop IPF, which is a severe, progressive fibrotic disease of pulmonary tissue with an unknown insult (Figure 1).

The lung tissue’s response to irritants or disease states can either result in resolution and return to normal function or the development of fibrosis that leads to a permanent reduction in total lung capacity. Here, we explore the possible interactions that favor the development of fibrosis. In this paper, we focus on Alveolar epithelial cells (AECs), various cytokines, transforming growth factor (TGF)-*β*, myofibroblasts, and environmental factors as they relate to the development of fibrosis post-COVID-19 infection.

AECs are an important group of cells within the parenchyma of the lungs that are responsible for many aspects of inflammatory regulation. AECs have two subtypes, AEC type I (AEC1) and AEC type II (AEC2), with both different roles and functions. AEC1 forms a simple squamous layer that normally surrounds the alveolar airspace and is involved primarily in gas exchange. The role of AEC2 cells is more complex. AEC2 cells can produce surfactant, a protein–lipid secretion that lowers the surface tension forces acting on the alveolus. AEC2 cells also serve as the predominant epithelial progenitor cells that are capable of differentiation into AEC1 cells. Loss of function or reduced AEC cell counts can lead to improper repair of the lung parenchyma, which can then lead to fibrosis [6].

The development of PF may be driven by microenvironmental insults to AECs. These microenvironmental insults can be from a variety of intrinsic (genetic, age, gender) or extrinsic factors such as infection (COVID-19), cigarette smoke, or occupational exposure. After the initial insult, the AEC2 cells may stimulate an immune response to clear or contain the microenvironmental insult. Optimally, once the insults are cleared, the inflammatory response subsides and the remaining AEC2s proliferate to repair damage to the tissue, ultimately leading to resolution. However, if the insults lead to the dysfunction of the AEC2, the cells will stimulate the recruitment of myofibroblasts. Myofibroblasts will lead to the activation of collagen synthesis, apoptosis of AEC2 cells, and ultimately, PF [7]. This dysregulation of the healing response and dysfunction of AEC2 cells is associated with long-lasting or recurring insults. The longer the insults take hold within the lung, the less likely a complete resolution will take place. Minimizing the offending insult quickly will play a key role in the prevention of fibrosis.

In addition to the AECs within the pulmonary tissue, many cytokines are involved in the upregulation of the inflammation that ultimately leads to fibrosis. The role of inflammatory cytokines in the progression of the PF is complex. The major stimulators include TGF-*β*. The role of TGF-*β* in PF is widespread, with many different mediators. A major cellular source of TGF-*β* is alveolar macrophages, bronchial epithelium, and hyperplastic AEC2 in response to inflammation and alveolar damage [8]. A primary role of TGF-*β* is to modulate the extracellular matrix (ECM) deposition through the activation of myofibroblasts. TGF-*β* seems to activate P120-catenin, an adhesion protein, to increase the fibroblastic foci and primary fibroblast in Bleomycin-challenged mice [9]. Bleomycin is a drug that is known to induce pulmonary fibrosis. Fibroblastic foci are a characteristic of idiopathic PF which indicate an area of lung destruction. TGF-*β*1, a class of TGF-*β*, regulates cell recruitment to sites of injury, induces fibroblast differentiation to myofibroblast, stimulates ECM production by myofibroblast, and inhibits the ECM degradation by matrix metalloproteinase [10]. There are also increased alveolar secretions of fibroblast growth factor (FGF)-2 in response to an increase in TGF-*β*1 activity through the activation of the extracellular-signal-regulated kinase (ERK) pathway. TGF-*β*1 response through the ERK kinase pathway and treatment with ERK1/2 inhibitors seems to attenuate the fibrotic differentiation [11]. Finally, TGF-*β* is regulated through the WNT/B-catenin pathway, and WNT10 overexpression has been shown to increase exacerbation and poor prognosis among IPF patients [12]. Given the many pro-fibrotic effects of TGF-*β* in the lung, targeting of TGF-*β* is theorized to be a desirable target for therapy in hope of preventing the progression of fibrosis.

Interestingly, there is a shift in the local cytokine profile in PF patients in addition to increased TGF-*β* signaling. A shift from a T_H_2 inflammatory profile to a T_H_1 inflammatory profile seems to decrease the progression of fibrosis [13]. The T_H_1 inflammatory profile includes the interferon-gamma (IFN)-γ and Interleukin (IL)-12; both attenuate the fibrosis whereas the T_H_2 inflammatory profile includes the IL-4, IL-5, and IL-13, which have been linked to fibrogenesis [14]. Other cytokines such as Tumor necrosis factor (TNF)-α, Platelet-derived growth factor (PDGF), CXC Chemokines, IL-1a, and TGF-*β* have been linked to the development of IPF [15]. The implication of the cytokines in IPF has prompted more investigation of different immunotherapies.

In addition to the cytokine response, the lung response to insult or injury includes the promotion of myofibroblasts. Myofibroblasts are differentiated from fibroblasts and then quickly recruited to the lung to facilitate the repair processes. Myofibroblasts migrate and secrete ECM within the area of damage, which provides a fundamental scaffolding for the repair process. The deposition of ECM can also serve as protective scar tissue by walling off the area of damage and preventing the spread of inflammation to the healthy areas. However, an imbalance in the ECM deposition leads to the formation of fibrosis. The stiffening of the ECM has been associated with the progression of IPF [16]. The regulation of myofibroblast activation and proliferation is a complex procedure and is currently under investigation. It is known that there are many factors facilitating its differentiation, including TGF- *β*1 through the microRNA-133a [17], or through Src family kinase [18]. Attenuating the differentiation of myofibroblast and limiting the collagen deposition may be a good strategy to slow the progression of pulmonary fibrosis.

Among others, oxidative stress is also implicated in the development of fibrosis. Reactive oxygen species (ROS) can be formed in excess from the innate inflammatory processes, such as in neutrophil granules, or environmental exposure. The lungs are especially vulnerable to oxidative stress, as the level of O2 is much higher than the rest of the body. High-level ROS is theorized to induce epithelial apoptosis, increase secretion of profibrotic cytokine, and increase differentiation of fibroblast to myofibroblast [19]. Of note, severe COVID-19 patients often need the ventilators to assist with breathing, which can increase the risk of oxidative stress in these patients.

COVID-19 patients also have an increased risk of pulmonary embolism (PE), further exacerbating the already decreased lung function. Patients with COVID-19 have an overall 16.5% incidence rate of developing a PE [20]. The hypercoagulability state that these patients are in predisposed them to developing chronic PE, which could also exacerbate the progression of PF.

## 3. COVID-19 and Lung Fibrosis

COVID-19 has quickly swept the globe and become the most pressing matter for many public health officials. The rate of transmission, the wide variety of presentations, and the novelty of the disease make it a very difficult disease to treat. As the treatment experience of severe COVID-19 patient cases increased, patient outcomes improved significantly. However, the long-term consequences of severe infection remain a big concern. One such long-term consequence is the development of PF. A minority of patients who develop COVID-19 subsequently develop hypoxemia despite receiving supportive care [21]. In the following section, the mechanisms behind COVID-19 infection and post-infectious fibrosis will be discussed.

COVID-19 is caused by the virus SARS-CoV-2 (acute respiratory syndrome coronavirus 2). SARS-CoV-2 is primarily transmitted via respiratory droplets; although the fecal-oral route has been shown, this represents a small minority of infections. SARS-CoV-2 is in a family of coronaviruses, which are enveloped, segmented RNA viruses [22]. The SARS-CoV-2 protein is cleaved by many serine proteases, which contribute to the rapid rate of transmission and infectivity. Upon binding to a host protein, either through the upper airway or the lower airway, a host protease, transmembrane serine protease (TMRSS)-2, reveals the fusion domain of the spike protein allowing attachment to the angiotensin-converting enzyme-2 (ACE2) on the receptor cell [23]. Following fusion, SARS-CoV-2 takes over the replication machinery of the host cell and then eventually releases the viral particles to infect neighboring cells, such as lung epithelial cells. It may render the SARS-CoV-2 infected diseases more challenging to treat.

Due to the nature of the attachment protein ACE2, SARS-CoV-2 potentially infects AEC2 cells, alveolar macrophages, enterocytes in the intestines, and possibly the basal epithelial cells in the nasal passages [19]. Since the AEC2 and alveolar macrophages are important regulators in promoting pulmonary fibrosis, COVID-19 infection may lead to PF as a long-term sequela.

Additionally, COVID-19 patients have increased levels of IFN-γ, TGF-*β*, IL-17, and IL-8 when compared to a normal control [24]. The increased level of IFN-γ suggests an increase in the Th1 response and a possible profibrotic inflammatory profile. The increases in the TGF-*β* also suggest the increased recruitment of fibroblast and the differentiation of myofibroblast. The increase in IL-17, which is a neutrophil attractant, can result in increased neutrophil degranulation, increasing oxidative stress in the lung, and promoting a fibrotic deposition [25]. These are conditions that favor the development of fibrosis in IPF as well, and they might have a role in the longer-term development of PF related to COVID-19 (Figure 2).

In addition to a similar pathophysiological microbiome within the insulted pulmonary tissue between IPF and COVID-19 patients, there is an overlapping patient demographic. The COVID-19 mortality rate is highest among the population of adults 85 years or older ranging from 10 to 27%, followed by patients between the age of 65–84 years old with a mortality rate of 3–11% [26]. Patients who are 65 years or older that are associated with a higher risk of poor outcome with a COVID-19 infection are correlated with the population that develops IPF, which is also age 65 or older [27]. This may mean that a subset of patients who encounter COVID-19 might have a predisposition to develop IPF, and the viral infection exacerbates the disease. On another note, an earlier study in the pandemic, looking at the outcome of patients with COVID-19, found that out of 278 pooled patients for the study, 72 patients (25.9%) developed COVID-19 pneumonia that required an intensive care unit (ICU) admission, 56 patients (20.1%) developed ARDS, 23 patients (8.3%) required mechanical ventilation and 9 (3.2%) required extracorporeal membrane oxygenation for refractory hypoxemia [28]. This study shows that COVID-19 infection has a high likelihood of developing complications. The severity of the disease tends to correlate with a greater rate of pulmonary fibrosis. A study looking at patients with Middle East respiratory syndrome, a related viral disease to COVID-19, showed that patients who developed PF correlated to the length of stay in the ICU [29]. This implicates the role of AECs in the progression of pulmonary fibrosis. The persistent, longer-term damage potentially disrupts the AECs to a point they cannot promote proper healing of the lung tissue. Furthermore, the assistance of mechanical ventilation can contribute to injury via the increase in ROS or via the biomechanical injury from the ventilation itself [30]. Additionally, lungs in the older population may not be as adaptive to insults as the younger lungs, leading to a more severe long-term outcome for these populations. Proper management, along with preventative measures, can be highly effective in protecting this population from long-term development of fibrosis from COVID-19.

## 4. The Potential Prevention of Lung Fibrosis

As discussed above, acute respiratory disease can lead to fibrosis of the lungs, a crippling with a poor prognosis. As of 2019, it has been reported that SARS-CoV-2 also causes acute disease in the lungs that lead to lung fibrosis. Above all, the rapid spread of the virus and the evolution of the outbreak into a pandemic in late 2019 and 2020 make treatment and guidelines against COVID-induced lung fibrosis all the more important [31].

Based on data collected by Chinese researchers from December 2019 through January 2020, the SARS-CoV-2 pandemic, severe cases of COVID-19 and the lung fibrosis that resulted from it are most similar to IPF [32]. Thus, current guidelines to palliate IPF may be viable options to treat COVID-induced fibrosis. Additionally, the mechanism of disease caused by COVID-19 involves a cytokine storm, severe inflammation, oxidative stress, and reactive oxygen species that lead to the development of ARDS. Thus, the proposed pharmacological therapies for SARS-CoV-2 should include both anti-fibrotic and anti-inflammatory effects [33]. In the following discussion, we will go over these potential therapeutic options for COVID-induced lung fibrosis, including pirfenidone and nintedanib, both of which have been approved for treatment of IPF [34,35].

Due to its anti-inflammatory and anti-fibrotic effects, pirfenidone may be used to treat COVID-induced lung fibrosis. The drug has been proven to inhibit TNF-alpha secretion in addition to other cytokines that cause the cytokine storm seen in SARS-CoV-2 patients [33]. Pirfenidone has also shown to decrease inflammatory response through inhibition of NLR family pyrin domain-containing (NLRP) 3, preventing the creation of an inflammasome complex that exacerbates COVID-induced inflammation [36]. In addition to its anti-inflammatory effects, pirfenidone also inhibits TGF-*β*-induced fibronectin synthesis as well as collagen 1 fibril formation [37,38]. Thus, this drug’s action on both inflammatory processes as well as the production of fibrotic tissue makes it a viable option for COVID-19 treatment and lung fibrosis prevention. A clinical trial by Zhang et al. at multiple centers in China looked at the outcomes of patients who received Pirfenidone along with standard treatment, when compared to the standard treatment alone [39]. The study concluded that pirfenidone did not reduce lung damage from COVID-19, but it did improve the cytokine storms and the coagulopathies that could possibly lead to long-term COVID complications. Furthermore, both groups had similar side effects, with diarrhea being the most common for the pirfenidone group.

As of 9 March 2020, nintedanib has been approved by the Food and Drug Administration (FDA) as a potential treatment option for progressive interstitial lung fibrosis. In 2021, a study of nintedanib for use against COVID-19 was conducted to see if the drug’s effects hold true against fibrotic disease caused by the virus. In this study by Umemura Y et al. [40], a randomized control trial was conducted to observe the effects of this drug. Although the results showed no significant difference in mortality between the two groups (23.3% mortality in control versus 20% mortality rate in the nintedanib group), the number of ventilator-free days was significant, with the nintedanib group spending fewer days on average on ventilator support compared to the control group. This study indicates the potential for nintedanib in improving outcomes of COVID-19 patients, and this drug could be considered as a therapeutic option for prevention of COVID-induced lung fibrosis through its mechanism of reducing the activation of fibrotic factor. Even though there were no differences in mortality, the reduction in days on the ventilators meant a decrease in inflammation, which can possibly decrease long-term complications of PF. Further investigation of application of nintedanib in pulmonary fibrosis patients will be required to determine if its presumed efficacy holds true in this potentially similar patient population.

Another pharmaceutical agent that may prevent COVID-induced lung fibrosis is azithromycin [41]. Azithromycin is an antibiotic that is in the macrolide family. It covers a wide variety of bacteria; however, it has been proven to exhibit antiviral effects as well [41]. Additionally, the macrolide also has immunomodulatory activity [42]. For these two reasons, azithromycin may be considered as a pharmacologic treatment to use against a SARS-CoV-2 infection.

Azithromycin has also been proposed to have a specific antiviral effect against COVID-19 and the proposed mechanism is complex. The macrolide has been proposed to act as an antiviral by both impairing lysosomal activity and preventing SARS-CoV-2 binding and entry into pulmonary interstitial cells [41]. However, two recent clinical trials with azythromcyin show that it does not improve outcome in anyway, and it should be strictly used when there is an indication for antibiotic use [42,43].

Histone deacetylase inhibitors may also have some therapeutic value for the treatment of SARS-CoV-2-induced fibrosis [44]. As was described in the previous sections, lung fibrosis is induced by the expression of TGF-ß, which is regulated by histone deacetylase activity [45]. Particularly, increased activity of histone deacetylase has been reported to promote the activity of TGF-ß, which then leads to further collagen synthesis and a worsening of interstitial fibrosis [46]. Thus, the anti-fibrotic action of histone deacetylase inhibitors is of note and needs further review.

Akin to histone deacetylase inhibitors, Biochanin A (isoflavone) is also thought to target TGF-ß-induced fibrosis, and thus, it should be considered as a potential therapeutic for COVID-induced lung fibrosis. In an in vivo study conducted by Sai Balaji Andugulapati et al., [46] two cohorts of mice were given bleomycin to induce pulmonary fibrosis. One of the two groups received Biochanin A and the other was the control. After the experiment, Andugulapati reported a significant decrease in TGF-ß expression, inflammatory marker expression, and collagen deposition in the lungs of the mice treated with Biochanin A. This suggests a potential role of this drug in humans with lung fibrosis. Of note, further investigation of its efficacy is required in higher animal models.

To capture a wide breadth of therapeutics for COVID-induced lung fibrosis, non-pharmacological therapeutics such as antioxidant and anti-inflammatory foods, including but not limited to vitamin D, may also be considered as adjuvant treatment options [47]. For instance, vitamin D has been reported to show antiviral effects in cells infected by herpes simplex virus (HSV), influenza, HCV, and even human immunodeficiency virus (HIV) [48]. In the lungs, vitamin D has been shown to reduce the activity of inflammatory cells such as histocytes and T-cells, which in turn reduces the amount of cytokines [49]. Additionally, patients who receive vitamin D supplementation levels have been associated with a better prognosis and have a higher chance of not acquiring acute respiratory disease compared to control groups [50,51,52].

## 5. Summary

Pulmonary fibrosis (PF) is a common disease state that remains a challenge to treat in the clinic. Unfortunately, patients who survive severe COVID-19 infection may suffer from a high incidence of PF. The formation of PF secondary to COVID infection has a similar underlying process to other pulmonary diseases. Although the pathways underlying fibrous tissue formation in PF are complex with many fibrotic factors involved, the primary mechanism is through the activation of the TGF-β1 signaling pathway. This article reviews the current understanding of the pathophysiological mechanisms underlying PF and therapeutic strategies. We also discussed the potential for preventing progression to the fibrotic state within the context of the COVID-19 pandemic.

## Figures and Tables

**Figure 1 cells-11-02489-f001:**
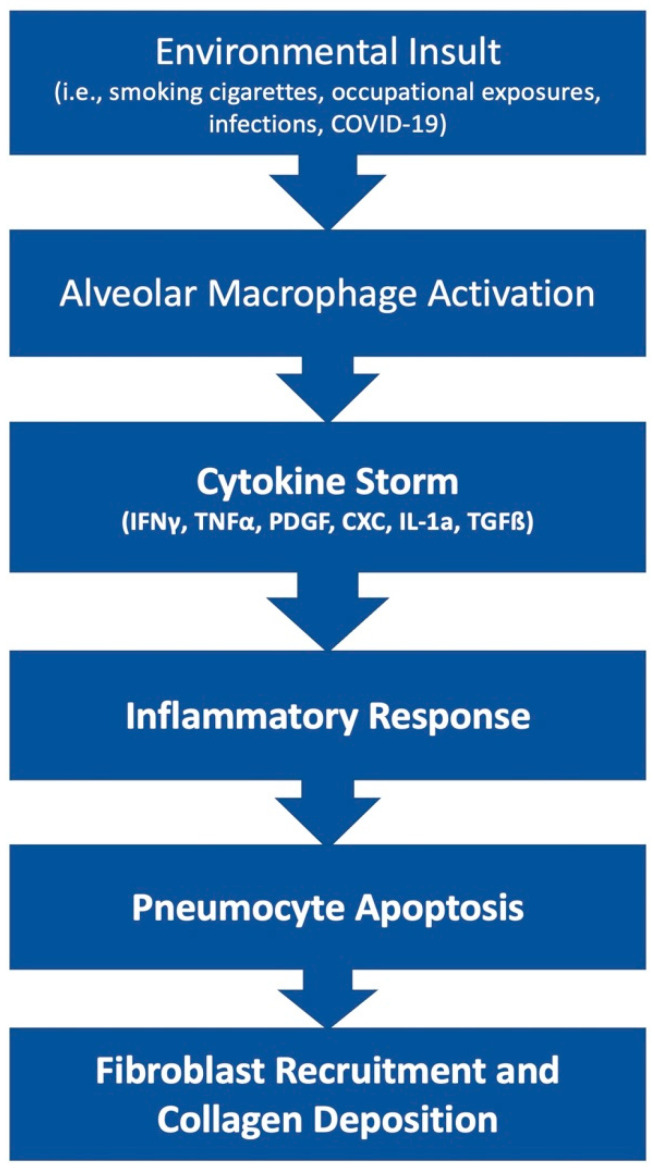
Simplified diagram of the development of fibrosis.

**Figure 2 cells-11-02489-f002:**
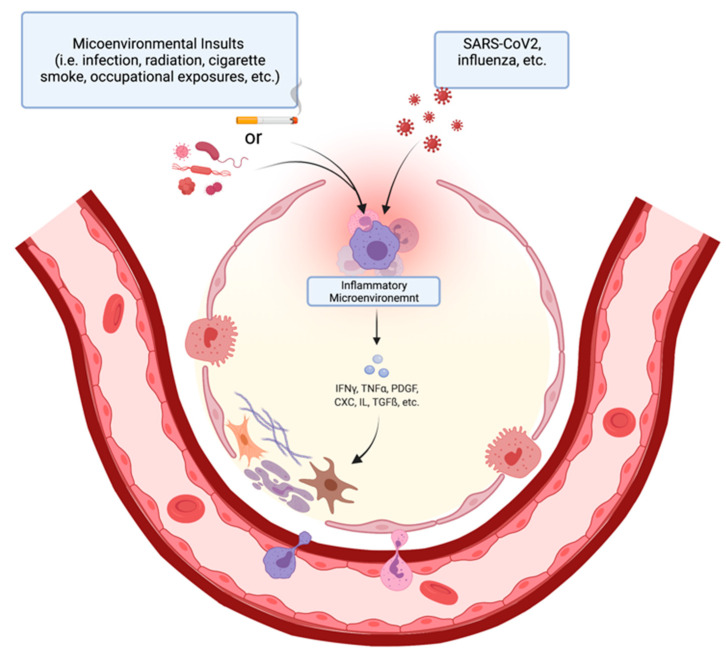
The onset and mechanism of fibrotic change in alveolar epithelium post microenvironmental insult (**left**). Immune cells (purple, pink) in the inflammatory microenvironment secrete pro-fibrotic factors that stimulate collagen deposition by fibroblasts (brown). Flow diagram of steps of pneumocyte apoptosis and fibrosis initiation (**right**). (Created with BioRender.com, accessed on 1 April 2022).

## Data Availability

Not applicable.

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
