# Peer review of "Pathophysiology of Pulmonary Fibrosis in the Context of COVID-19 and Implications for Treatment: A Narrative Review"

_cells, 2022, doi:10.3390/cells11162489_

Round 1

Reviewer 1 Report

The manuscript entitled “Pathophysiology of pulmonary fibrosis in the context of COVID-19 and implications for treatment” by Tran, et al describes the development of pulmonary fibrosis after COVID-19 infection and the findings suggesting the underlying pathophysiology of this condition.  Attempts are made to tie in the findings of pulmonary fibrosis seen in idiopathic pulmonary fibrosis (IPF) to that seen after COVID-19 infection.  However, a description of post-COVID-19 pulmonary fibrosis and its underlying pathophysiology is premature due to the lack of extensive clinical data and specific animal models to identify involved pathways.  As such the manuscript is mostly speculative.   This reviewer has the following concerns regarding the manuscript.

Major concerns:

1.  The authors state that there are several similarities between patients with pulmonary fibrosis after COVID-19 infection and IPF but the similarities listed are non-specific and are generalizable to patients with critical illness with and without pulmonary fibrosis.  Symptoms of shortness of breath and restrictive physiology are not unique among pulmonary fibrosis and mortality is higher across all spectrum of lung disease with advanced age, particularly in patients who are critically ill.  In all conditions associated with prolonged use of mechanical ventilation in the intensive care setting development of pulmonary fibrosis is a risk factor.

2. The underlying pathophysiology for the development of pulmonary fibrosis in the setting of COVID-19 is very early in investigation without significant basic science exploration undertaken.  The manuscript speculates that the pathophysiology for COVID-19 induced pulmonary fibrosis is similar to IPF but no conclusive studies have been performed that directly identify a connection and reliance on clinical data alone does not allow for accurate correlations.

3. The discussion related to use of nintedanib in COVID-19 outlines its use in a clinical trial to assess its effect on acute therapy and critical care outcomes during acute COVID-19 infection and was not designed to explore to post-infection incidence of, and potential treatment of, pulmonary fibrosis, the focus of this manuscript.

4. The discussion regarding the potential use of Biochanin A in pulmonary fibrosis is very premature with no studies performed in models of COVID-19 induced pulmonary fibrosis or other viral induced respiratory illnesses.  The bleomycin model does not fully replicate the pathophysiology of IPF and although is a well established model, it is a poor model for IPF.

Author Response

Major concerns:

  1. The authors state that there are several similarities between patients with pulmonary fibrosis after COVID-19 infection and IPF but the similarities listed are non-specific and are generalizable to patients with critical illness with and without pulmonary fibrosis.  Symptoms of shortness of breath and restrictive physiology are not unique among pulmonary fibrosis and mortality is higher across all spectrum of lung disease with advanced age, particularly in patients who are critically ill.  In all conditions associated with prolonged use of mechanical ventilation in the intensive care setting development of pulmonary fibrosis is a risk factor.

Answer -  The purpose of this article was not to compare the acute infection of COVID-19 to pulmonary fibrosis, but rather elucidating the possible long-term complication of pulmonary fibrosis from an acute infection of COVID-19. This was done by drawing similarity between the pathophysiology of IPF, as well as other viral infection such as SARS to suggest how PF might arise from an acute COVID-19 infection.

  1. The underlying pathophysiology for the development of pulmonary fibrosis in the setting of COVID-19 is very early in investigation without significant basic science exploration undertaken.  The manuscript speculates that the pathophysiology for COVID-19 induced pulmonary fibrosis is similar to IPF but no conclusive studies have been performed that directly identify a connection and reliance on clinical data alone does not allow for accurate correlations.

Answer - It is true that there have been no conclusive studies that there has been no studies that explore the pathophysiology of COVID-19 induced pulmonary fibrosis, this paper looks to attempt to guide possible mechanism of such process based our research experience and knowledge in fibrosis study. The comparison to IPF merely present as a way to explain the process of PF, as it is the most common, as well as the most studied PF process.

  1. The discussion related to use of nintedanib in COVID-19 outlines its use in a clinical trial to assess its effect on acute therapy and critical care outcomes during acute COVID-19 infection and was not designed to explore to post-infection incidence of, and potential treatment of, pulmonary fibrosis, the focus of this manuscript.

Answer - The current understanding is to reduce the length on the ventilator and hospital stay should reduce the inflammatory changes to the lungs, which will decrease the remodeling that mentioned in the article, and hopeful shows a possible reducing in PF. Revision has been made to make this point more clear.

  1. The discussion regarding the potential use of Biochanin A in pulmonary fibrosis is very premature with no studies performed in models of COVID-19 induced pulmonary fibrosis or other viral induced respiratory illnesses.  The bleomycin model does not fully replicate the pathophysiology of IPF and although is a well-established model, it is a poor model for IPF.

Answer - This is true that Biochanin A has not been tested in in models of COVID-19 induced PF, but this is a possible treatment that has not yet explore. Given the lack of algorithm for the prevention of pulmonary fibrosis, we feel that this is a good possibility that has not been explored.

Reviewer 2 Report

In this review the authors summarize what is known so far about the pathophysiology of pulmonary fibrosis following COVID infection. The topic is of interest and the discussion is simple enough to follow even for non-expert readers of the subject.

COMMENTS

- Please, clarify in the title what type of revision it is (narrative or systemic)

- The abstract is quite repetitive. in the case of systematic revision, the criteria for inclusion of the papers considered must be set

- In the introduction, clearly state the objective of the review

- It would be useful to cite data about the occurrence or prevalence of pulmonary fibrosis in patients with long COVID

Author Response

In this review the authors summarize what is known so far about the pathophysiology of pulmonary fibrosis following COVID infection. The topic is of interest and the discussion is simple enough to follow even for non-expert readers of the subject.

COMMENTS

- Please, clarify in the title what type of revision it is (narrative or systemic)

Answer: Thanks for the suggestion, Edits were made to clarified in the title and the abstract.

- The abstract is quite repetitive. in the case of systematic revision, the criteria for inclusion of the papers considered must be set

Answer: We agree, and the revision was made for better flow and reduce repetitiveness

- In the introduction, clearly state the objective of the review

Answer: Thanks, Edits were made to clearly state the objective

- It would be useful to cite data about the occurrence or prevalence of pulmonary fibrosis in patients with long COVID

Answer: Thanks for the suggestion, we agree with this point; however, we did not find any data about the prevalence or occurrence of pulmonary fibrosis as this is an area under active investigation.

Reviewer 3 Report

This is an interesting review of the development and potential therapeutic interventions of pulmonary fibrosis in the context of COVID-19. Given the fact that COVID-19 has practically become endemic, it is important to address potential long-term consequences of the disease. This review brings out some of the pulmonary consequences of COVID 19. However, the authors limit their observations and define the mechanisms of COVID-19-induced PF only at the lung level while the vascular involvement in COVID-19 consequences is well-established and plays an important role in this pathology.

As stated in this paper, the pathophysiology of PF indeed involves lung oxidative stress and inflammatory cells, however COVID 19 has also been shown to induce systemic inflammation and coagulopathy, which could contribute to the onset of lung injury. The lung is the largest microvascular bed in the body and changes in vasoreactivity and endothelial dysfunction causing increased vascular permeability, oxidative injury, and impaired gas exchange, are all implicated in the development of PF. A significant pulmonary consequence of COVID-19 is targeting the microvasculature. Coagulopathy and persistent vascular inflammation cause greater collagen production and increased recruitment of inflammatory cells by activated endothelial cells, greatly contributing to lung injury. This vascular component is thought to develop early in COVID19 and contribute to ARDS which will then cause fibrosis (Polak et al., Mod. Pathol. 2020, 33(11): 2128). The authors’ interpretation of the mechanism of COVID19-induced pulmonary fibrosis ignores the role of COVID coagulopathy and its consequences on lung microvascular dysfunction.

In the “potential prevention of lung fibrosis” paragraph, the authors discuss therapeutic approaches that have been successful at mitigating IPF. However, the discussion is somewhat partial as they do not fully present the results of the trials for these drugs in COVID patients, that showed no improvement in COVID-19 -induced damage. The Umemura study shows no significant difference between drug and placebo treated patients, except for fewer days of ventilation, so some improvement on lung function (not surprisingly) but not on mortality. These results are reported but no mention is made of the limited number of participants. Pirfenidone is indeed appropriate to reduce the inflammation, as described. However, a Chinese clinical trial, enrolling a large number of COVID patients, found no beneficial effect to reduce lung injury, but a significant reduction of the cytokines storm and coagulopathy (lower D-dimer levels) (Chinese Medical Journal, 2022, 135(3):368). The authors then discuss the potential therapeutic effect of Azithromycin but fail to mention two clinical trials that found no benefit in COVID 19 patients (Oldenburg et al., JAMA 2021, 326:490; and Alarger et al., The Lancet, 2021, 397 (10279: 1063). The discussion of vitamin D as an antiviral and granting protection from pulmonary complications is incomplete as well. While COVID 19 patients had significantly lower levels of vitamin D, it is not clear that vitamin D could be used as a treatment or that these lower levels are related to the severity of COVID19 symptoms and nothing shows that it could have any effect on PF (Weir et al., Cin. Med. (Lond), 2020, 20(4):e107.)

Figure 2: It would be helpful if the legend was more explicit and identify the cells represented by each color.

Author Response

This is an interesting review of the development and potential therapeutic interventions of pulmonary fibrosis in the context of COVID-19. Given the fact that COVID-19 has practically become endemic, it is important to address potential long-term consequences of the disease. This review brings out some of the pulmonary consequences of COVID 19. However, the authors limit their observations and define the mechanisms of COVID-19-induced PF only at the lung level while the vascular involvement in COVID-19 consequences is well-established and plays an important role in this pathology.

As stated in this paper, the pathophysiology of PF indeed involves lung oxidative stress and inflammatory cells, however COVID 19 has also been shown to induce systemic inflammation and coagulopathy, which could contribute to the onset of lung injury. The lung is the largest microvascular bed in the body and changes in vasoreactivity and endothelial dysfunction causing increased vascular permeability, oxidative injury, and impaired gas exchange, are all implicated in the development of PF. A significant pulmonary consequence of COVID-19 is targeting the microvasculature. Coagulopathy and persistent vascular inflammation cause greater collagen production and increased recruitment of inflammatory cells by activated endothelial cells, greatly contributing to lung injury. This vascular component is thought to develop early in COVID19 and contribute to ARDS which will then cause fibrosis (Polak et al., Mod. Pathol. 2020, 33(11): 2128). The authors’ interpretation of the mechanism of COVID19-induced pulmonary fibrosis ignores the role of COVID coagulopathy and its consequences on lung microvascular dysfunction.

Answer: We agree with reviewer, The edits was made to address the possible contribution of the hypercoagulability state and its impact on the outcome of pulmonary fibrosis.

In the “potential prevention of lung fibrosis” paragraph, the authors discuss therapeutic approaches that have been successful at mitigating IPF. However, the discussion is somewhat partial as they do not fully present the results of the trials for these drugs in COVID patients, that showed no improvement in COVID-19 -induced damage. The Umemura study shows no significant difference between drug and placebo treated patients, except for fewer days of ventilation, so some improvement on lung function (not surprisingly) but not on mortality. These results are reported but no mention is made of the limited number of participants. Pirfenidone is indeed appropriate to reduce the inflammation, as described. However, a Chinese clinical trial, enrolling a large number of COVID patients, found no beneficial effect to reduce lung injury, but a significant reduction of the cytokines storm and coagulopathy (lower D-dimer levels) (Chinese Medical Journal, 2022, 135(3):368). The authors then discuss the potential therapeutic effect of Azithromycin but fail to mention two clinical trials that found no benefit in COVID 19 patients (Oldenburg et al., JAMA 2021, 326:490; and Alarger et al., The Lancet, 2021, 397 (10279: 1063). The discussion of vitamin D as an antiviral and granting protection from pulmonary complications is incomplete as well. While COVID 19 patients had significantly lower levels of vitamin D, it is not clear that vitamin D could be used as a treatment or that these lower levels are related to the severity of COVID19 symptoms and nothing shows that it could have any effect on PF (Weir et al., Cin. Med. (Lond), 2020, 20(4):e107.)

Answer: This is a good suggestion, however, many of these trials were on going during the reviews of this article. We have completed the edits that was made to include the results of the clinical trials. However, the Umemura study and the Chinese clinical trials were only looking at the reduction of pulmonary fibrosis, we won’t include to discuss in this review.

Figure 2: It would be helpful if the legend was more explicit and identify the cells represented by each color.

Answer: Thanks, we have made the edited with legends for more clarity.

Reviewer 4 Report

The manuscript is interesting and well written. However, I believe that for a completeness of information it would be necessary to integrate with a paragraph concerning the immune cells that play a role in the fibrosis process, such as neutrophils and macrophages.

Furthermore, considering that the authors also briefly mention the possible role of miRNAs, it would be interesting to add a small discussion on the recently published works on this topic.

Finally, the authors very briefly describe the bleomycin animal model. It would be interesting to deepen the animal models of fibrosis.

References seems uneven, I recommend a review

Author Response

Dear editor,

Please find a copy of the response letter, in which we have answered all questions point-by-point. If any other information is needed, please let us know. Thank you.

Bests,

Yong Li MD, PhD

Reviewer 4:

The manuscript is interesting and well written. However, I believe that for a completeness of information it would be necessary to integrate with a paragraph concerning the immune cells that play a role in the fibrosis process, such as neutrophils and macrophages.

Answer: we agree with the reviewer, however, we opted to not further elaborate on the role of immune cells such as neutrophils and macrophages as we feel that be not as beneficial since we’re discussing a viral eitology of PF.

Furthermore, considering that the authors also briefly mention the possible role of miRNAs, it would be interesting to add a small discussion on the recently published works on this topic.

Answer: Yes, we have added a small discussion about the current published on of miRNAs.

Finally, the authors very briefly describe the bleomycin animal model. It would be interesting to deepen the animal models of fibrosis.

Answer: Although we agree this is good suggestion, A deeper understanding of animal models of fibrosis would be not as beneficial to the article and it might be detracting from initial objective. Thanks for this suggestion.

References seems uneven, I recommend a review

Answer: yes, we have corrected those references.

Round 2

Reviewer 1 Report

The revised manuscript is not significantly different than the original submission and does not adequately address the reviewer's concerns of the original submission.  The manuscript proposes to discuss the underlying pathophysiology of post-COVID pulmonary fibrosis but as little is presently understood there is little to discus apart from speculation of specific factors regulating lung fibrosis after COVID infection.  In addition, proposed treatments are only speculative and although analogy is made to use of treatments utilized in the management of IPF, these therapies may improve physiologic clinical results but their role on directly decreasing lung fibrosis is not fully known.

Reviewer 3 Report

The authors have adequately addressed the critiques

Reviewer 4 Report

I believe that the authors have partially improved the manuscript according to the indications and that this version is acceptable for publication